# 3DPhenoMVS: A Low-Cost 3D Tomato Phenotyping Pipeline Using 3D Reconstruction Point Cloud Based on Multiview Images

Yinghua Wang, Songtao Hu, He Ren, Wanneng Yang and Ruifang Zhai *

National Key Laboratory of Crop Genetic Improvement, National Center of Plant Gene Research (Wuhan), College of Informatics, Huazhong Agricultural University, Wuhan 430070, China
* Correspondence: rfzhai@mail.hzau.edu.cn; Tel.: +86-13628627629

**Abstract:** Manual phenotyping of tomato plants is time consuming and labor intensive. Due to the lack of low-cost and open-access 3D phenotyping tools, the dynamic 3D growth of tomato plants during all growth stages has not been fully explored. In this study, based on the 3D structural data points generated by employing structures from motion algorithms on multiple-view images, we proposed a 3D phenotyping pipeline, 3DPhenoMVS, to calculate 17 phenotypic traits of tomato plants covering the whole life cycle. Among all the phenotypic traits, six of them were used for accuracy evaluation because the true values can be generated by manual measurements, and the results showed that the $R^2$ values between the phenotypic traits and the manual ones ranged from 0.72 to 0.97. In addition, to investigate the environmental influence on tomato plant growth and yield in the greenhouse, eight tomato plants were chosen and phenotyped during seven growth stages according to different light intensities, temperatures, and humidities. The results showed that stronger light intensity and moderate temperature and humidity contribute to a higher biomass and higher yield. In conclusion, we developed a low-cost and open-access 3D phenotyping pipeline for tomato and other plants, and the generalization test was also complemented on other six species, which demonstrated that the proposed pipeline will benefit plant breeding, cultivation research, and functional genomics in the future.

**Keywords:** 3D phenotyping; 3D reconstructed point cloud; structure from motion; growth analysis; whole growth stages; tomato

## 1. Introduction

The tomato, as the most popular vegetable crop, is widely cultivated worldwide under outdoor and indoor conditions due to its high nutrition and health benefits [1]. It is essential to estimate the phenotypic traits of tomato plants and explore phenotypic trait variation during different growth stages, which can help to address and understand the relation between tomato plant growth and the surrounding environment. Although great advances have been made in tomato breeding, more efforts should be made to characterize objective, reliable, and informative measurements of phenotypic traits to push breeding further [2]. Thus, a sustainable breakthrough in tomato phenotyping is urgently needed.

In the past decade, various phenotyping technologies have drawn much attention in the agricultural field because of the rapid development of new sensors and correspondingly high automation technology in the urgent need of non-destructivity [3]. Two-dimensional imaging technologies have been applied in structural trait estimation, growth analysis, and yield estimation at different time points [4–7]. Although many image processing and analysis algorithms have been developed on various crops, several defects cannot be avoided, such as the ambiguity of plant size caused by camera viewpoints and camera-object distance, the lack of 3D information regarding plant volume, and self-occlusion problems caused by the complex structure of plants [6].

Thus, 3D phenotypic techniques have gradually become a powerful tool for obtaining phenotypic traits due to their noninvasive and noncontact properties and in particular their advantages in obtaining 3D geometric structural information compared to conventional 2D technologies [6]. The superiority of using 3D information in the calculation of phenotypic traits, such as plant volume, plant height, and leaf length, has been demonstrated [8]. As 3D point clouds are becoming a standard data type for digitizing plant architecture in the laboratory or in the field, generating 3D points of crop plants is the essential challenge encountered in 3D phenotyping that needs to be addressed first. The most commonly used technologies for generating 3D points can be divided into two categories, namely, passive and active solutions [9]. For the last decade, 3D point clouds of plants have been able to be created by active sensors, such as in laser scanning. Laser scanning often outputs a complete and accurate 3D point cloud of one plant or plants at field scale [10]; however, the high cost of laser scanners is the main bottleneck that limits their wide application. Compared to laser scanning technologies, passive reconstruction methods, usually camera-based, are still the most competitive and widely applicable technologies for reconstruction solutions because only one or multiple easily accessible and affordable camera is needed.

Generating the 3D structure of objects from a series of 2D images is the first task that needs to be addressed before extracting 3D phenotypic traits. Multiview stereo (MVS) technology is the mainstream algorithm. Compared to binocular stereo vision, a method to derive 3D information based solely on the relative positions of the object in the two cameras, multiview stereo technology is more applicable because a complete point cloud of an object can be generated [11]. Data collection can be completed in two scenarios: a fixed-camera scenario, in which one or multiple cameras are fixed in the scene, and a moving-camera scenario, in which a camera moves around the object in the scene. In the fixed-camera scenario, the camera remains in one location, and auxiliary equipment, such as a turn stand, is needed to take multiple images of one object from different viewpoints. Alternatively, multiple cameras can be equipped at different viewpoints with respect to the object in order to create a complete representation of the object. The earlier reconstruction process for plants usually used this strategy. Calibration is the most important prerequisite work, followed by different reconstruction strategies, mainly correspondence-based methods [4,9,11–16] and carving-based methods [17–24].

In the moving-camera scenario, a camera is moved around an object to capture multiple images from different viewpoints. In this case, structure from motion (SfM) and MVS are combined together to generate a dense point cloud for objects, such as wheat and rice plant [25], Arabidopsis [12], tomato plants [11], Scindapsus and Pachira macrocarpa [11], maize plants [9,14], sweet potato plants [6], and even plant roots [26].

Phenotyping methods of tomatoes have also been studied in several literatures. In the conventional 2D imaging scenario, image analysis was mainly conducted on fruits and seeds [2]. In addition, unmanned aerial vehicle (UAV)-based imagery and random forests were used for biomass and yield prediction at the field scale [27]. Regarding the 3D reconstruction of tomato plants, virtual dynamic models of tomato development was presented using GREENLAB dual-scale automation [28] and a parametric L-System [29,30] at the organ level. To produce an accurate 3D model of tomato plant canopies, a close-range photogrammetric package, which is basically a correspondence-based method, was used to generate 3D points of large tomato plant canopies for volume estimation [31]. A correlation of $R^2$ values of 0.75 between the measured volumes and manually derived reference volumes was found, and the correlation of $R^2$ values between the leaf area index (LAI) and manual volume was found to be 0.82. A combined SfM and MVS technique was carried out in 2015 [8]; the phenotypic traits, including leaf area, main stem height, and convex hull, of the complete plant were estimated and compared to the reference data acquired by a laser scanner, and high $R^2$ values were found, greater than 0.9. In addition, Nguyen built a system by integrating five stereo camera pairs, a structured light system, and software algorithms to model plants [32], which was basically a correspondence-based approach. The results reached a recall of 0.97 and a precision of 0.89 for leaf detection and

less than a 13-mm error for plant size, leaf size, and internode distance. However, most of the previous algorithms mainly take the certain growth stages of plants into consideration. In other words, the potential of 3D phenotyping technologies has not been fully exploited on tomato plants over the full life cycle.

Thus, 3DPhenoMVS, a low-cost and open-access 3D tomato phenotyping and growth analysis pipeline using a 3D reconstruction point cloud based on multiview images, was proposed to address the abovementioned issues. In addition, with 3D traits, we investigated the environmental influence on tomato plant growth and yield in greenhouses.

## 2. Materials and Methods

### 2.1. Pipeline of 3DPhenoMVS

Considering the advantages of image-based reconstruction approaches, such as noninvasiveness, flexible data collection, and low cost, we present a spatial-temporal method called 3DPhenoMVS to reconstruct 3D point clouds of tomato plants; this method covers the full life cycle of tomato plants based on the SfM-MVS approach, uses multiple images captured by a consumer camera, and calculates 3D tomato phenotypic traits. The proposed 3DPhenoMVS consists of 12 modules, as shown in Figure 1 and listed as follows: (1) Environmental monitoring. A distribution map of environmental differences is illustrated for the tomato plant sample selection in Figure 1a. (2) Multiple-image data collection covering the whole life cycle of tomato plants, as shown in Figure 1b. (3) Three-dimensional point cloud generation through the combined SfM-MVS algorithms. (4) Alignment of point clouds regarding one tomato plant. For large tomato plants, image data are collected at two height levels, and the generated point clouds need to be aligned and registered together to produce a complete representation of the whole tomato plant. (5) Segmentation of stalk and leaf point clouds. (6) Skeletonization of stalk point clouds for structural phenotypic trait estimation. (7) Node detection on the skeletonized stem point clouds. Node detection is conducted on the skeletonized points; hence, the phenotypic traits, including node number, internode length, and stem length, are calculated automatically. (8) Three-dimensional structural phenotypic trait calculation and analysis. (9) Multiple-image data collection of tomato fruits. (10) Three-dimension point cloud generation through the combined SfM-MVS algorithms. (11) Alignment of point clouds regarding one tomato fruit. (12) Tomato fruit phenotypic trait calculation and analysis.

### 2.2. Environmental Conditions of Plant Material and Image Collection

Due to the superiorities of the image-based reconstruction method mentioned above, a consumer-cost digital camera (EOS 77D, Canon Corporation, Tokyo, Japan) was employed in our study. The tomato plants were planted in troughs in a form of soilless culture in a greenhouse that was approximately 400 m$^2$. Tomato seedlings were transplanted into the greenhouse on 1 August 2020, which is located in Liaocheng City, Shandong Province, China (Figure 2). The greenhouse was equipped with common management facilities, including reservoirs, drip irrigation and drainage components, draught fans, wet curtains, and vents. The reservoir and wet curtains were located at one end of the greenhouse, and the draught fan was located at the other end. There were two vents, upper and lower vents, allowing for air circulation. A total of 35 rows of plants were planted, with 55 plants in each row. The row spacing was 0.8 m, and the plant spacing was 0.16 m. The plant height was approximately 0.3~0.4 m, and the seedlings were hung 17 days after planting. The planting scenario is illustrated in Figure 2.

Ideally, the environmental conditions inside the greenhouse would be consistent; however, there were still some local microenvironment differences (microclimates) in the greenhouse. Hence, the local environmental parameters, including temperature, humidity, and light intensity, in the greenhouse were measured. The temperature and humidity were measured by a RENKE COS03 Temperature data logger (Shandong Renke Control Technology Co., Ltd., China), and the light intensity was measured by an EVERFINE PLA-20 plant lighting analyzer (EVERFINE Corporation, China). To avoid differences due

to manual intervention, 40 sample locations of environmental factors were first monitored and recorded evenly in the greenhouse. The measurement work lasted for 8 consecutive days. The change in the environmental parameters for the 40 sample locations is illustrated in the 3D scatter plots shown in Figure 2b.

As illustrated in Figure 2a, draught fans were equipped at the east end of the greenhouse, and wet curtains were equipped at the west end. In addition, the upper vents were mounted in the northern area, and the lower vents and drainage pipes were set in the southern area. All the conditional parameters were measured under the same controlled conditions, i.e., the vents were closed, and the draught fans and wet curtains were open. Due to the effect of the wet curtain and the draught fan, the closer to the wet curtain, the higher the humidity was, and the closer to the draught fan, the higher the temperature was. Similarly, drainage troughs would lead to higher humidity in closer areas, and vents would lead to lower temperatures. In addition, occlusion of the shelter would cause lower light intensity. A closer look at Figure 2b shows that gradual changes occurred for all three environmental parameters. Based on the discussion above, 8 tomato plants located in the 8 positions where environmental conditions obviously varied were selected as the target samples in this study, and they are illustrated by circles in Figure 2b.

For convenience of description, the notation "plant i-j" is used to represent one plant, where i represents the number of the row in which the tomato plant is located, which ranges from 1 to 35, and j represents the number of the column in which the tomato plant is located, which ranges from 1 to 55. The selected target samples were as follows: plants 1-1, 1-55, 10-53, 30-4, 30-52, 35-22, 10-3, and 20-28.

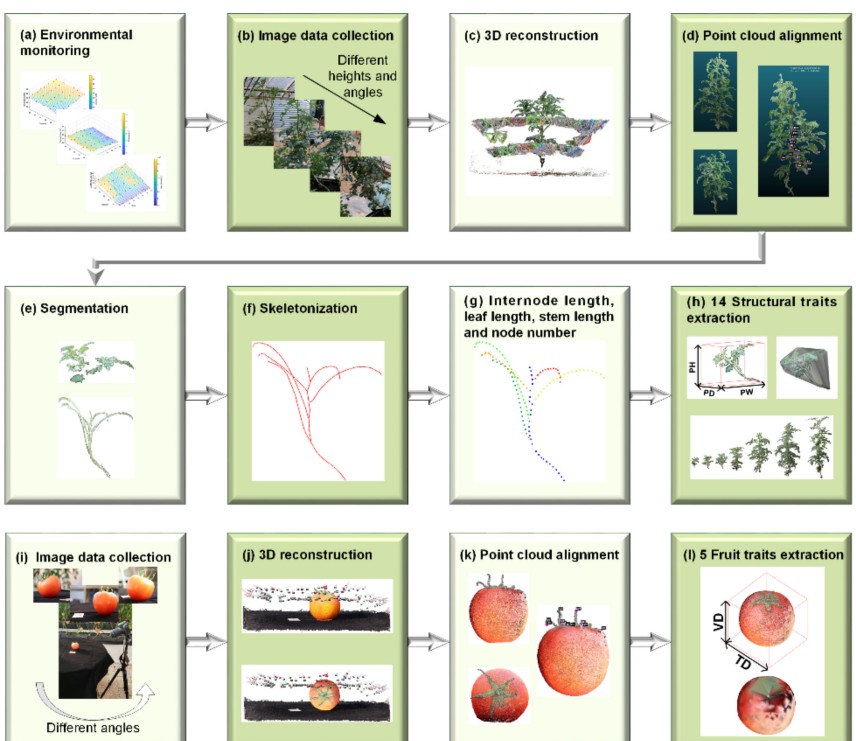

**Figure 1.** 3DPhenoMVS: A structural phenotypic trait estimation and growth analysis pipeline for tomato plants, covering the whole life cycle. (**a**) Environmental monitoring to determine the target plants, (**b**) image data collection to obtain multiview image sequences, (**c**) 3D reconstruction from images to generate 3D point clouds, (**d**) point cloud alignment to output a complete tomato plant model, (**e**) stalk and leaf point cloud segmentation to separate stalk point clouds and leaf (except petiole) point clouds, (**f**) skeletonization of stalk point clouds, (**g**) phenotypic trait extraction and visualization based on the skeletonization results, (**h**) extraction and analysis of structural phenotypic traits, (**i**) image data collection to obtain multiview image sequences, (**j**) 3D reconstruction, (**k**) point cloud alignment to obtain a complete tomato fruit model, (**l**) extraction of 5 fruit phenotypic traits.

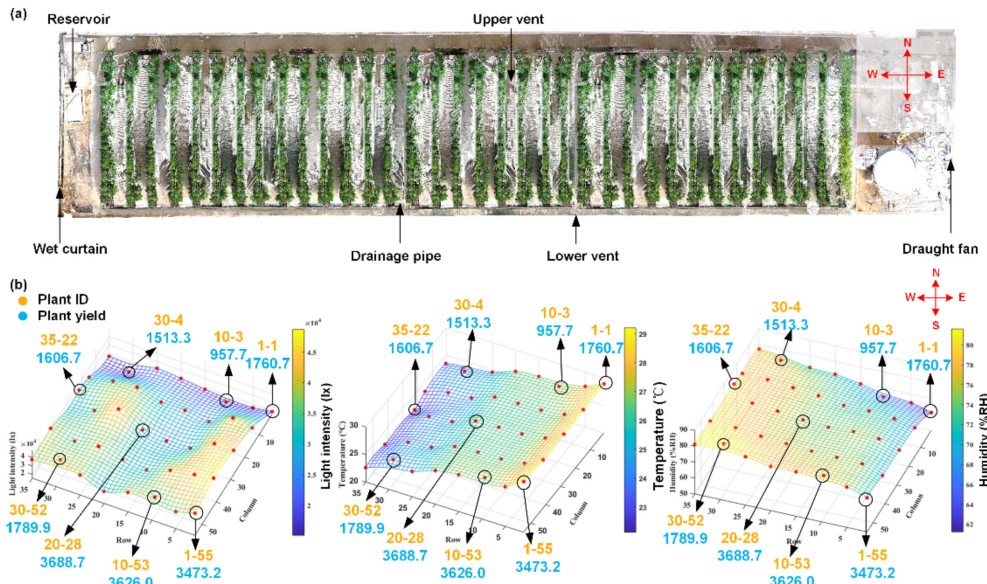

**Figure 2.** The plant scenario in the greenhouse. (**a**) Top view of the greenhouse and the locations of draught fans, reservoirs, wet curtains, drainage pipes, and vents. (**b**) The environmental parameters vary in different locations with light intensity changes (**left**), temperature changes (**middle**), and humidity changes (**right**). The yield for each individual tomato plant is also shown.

For each plant, the user moved the camera around the tomato plant in a circular fashion, and approximately 100 images were taken for each tomato plant. Image collection started on the 14th day after transplantation. When the tomato plants grew very quickly in the early stage, images were collected every 6 days. Since the growth rate of the tomato plants decreased in the later period, images were taken every 15 days until the tomato fruits were mature. As the plant grew, self-occlusion often occurred because the bottom parts could be occluded by the top parts. In this case, the digital camera needed to be set at different height levels, and two sets of images were collected in a circular fashion. When the tomato fruits were mature, fruits were randomly picked from each target individual plant for 3D phenotypic trait calculation. To produce a complete point cloud of one tomato fruit, the data collection was conducted from two different viewpoints. Generally, around 100 images were taken for one fruit. First, the fruit was placed on the table with the pedicle at the top. Then, it was placed with the pedicle at the front. Thus, two point clouds were generated regarding one plant, and the alignment was further carried out to create a complete point cloud regarding one tomato fruit. The data collection scenario and point cloud alignment are illustrated in Figure 1i–k.

### 2.3. Three-dimensional Reconstruction of Tomato Plants by SfM

SfM is a photogrammetric range imaging technique for estimating 3D structures from image sequences. It takes multiple images as input and outputs the camera parameters for each image as well as the coarse 3D shape of the object; that is, it performs sparse reconstruction. Usually, key point feature detection algorithms, such as the scale-invariant feature transform (SIFT) image feature descriptor [33] or variations of it, were implemented on the input images and later used for key point matching, and bundle block adjustment from the photogrammetry community was also introduced to estimate the accurate camera parameters for each image [34]. Through these procedures, the intrinsic and extrinsic parameters for each image were calculated properly, and the sparse points of tomato plants were also generated. The MVS algorithm then used the calibrated images to derive a very dense point cloud, which was color-coded using the original image data, by adopting epipolar geometry. The above procedures can be implemented with the free academic software VisualSFM [34,35]. Theoretically, all the images including the upper and lower

parts of tomato plants at the later growth stages can be used for 3D point cloud generation simultaneously by inputting all of them. In practice, due to the limitation of the computing power, if the number of images was too large, the computer would crash. Thus, the upper part and lower part were reconstructed separately. In cases where two sets of point clouds for the upper and lower parts of one tomato plant were generated, the complete point cloud could be created by aligning the two sets of point clouds by locating 3 or more correspondences between them.

### 2.4. Fruit Point Cloud Acquisition and Processing

In order to generate a complete point cloud regarding tomato fruit, image data collection was conducted from two viewpoints, and correspondently, two sets of point clouds consisting of fruit and tables were produced. First, the table points were removed while only the fruit points were kept. Then, the fruit point cloud with the pedicle at the top was set as the reference, and the other point cloud with the pedicle at the side was aligned together. After that, the normal was calculated by fitting the table points in the reference frame, and it was used to rotate the fruit point clouds into a reference frame with the vertical upward direction as the prime direction. Through these procedures, the direction of the vertical diameter was consistent with the vertical upward direction, and the distance between the maximum and minimum value along the prime direction was calculated as the vertical diameter. Meanwhile, the maximum distance along the projected area at the horizontal plane was set as the transverse diameter. The measurements of vertical diameter and transverse diameter are shown as Figure 1l, and the source code and detailed implementation are described in Supplementary Note S1.

### 2.5. L1-Medial Skeleton of the Stalk Point Cloud

Minor manual work was involved to separate the leaf points and stalk points, as shown in Figure 1e. Then, based on the stalk points, extracting a skeletal representation was an effective tool for geometric analysis and manipulation. In addition, a significant decrease in the time cost for the geometric analysis of the skeletal representation could be achieved compared with the analysis of the original point cloud. An *L1*-medial skeleton algorithm [36] was demonstrated to be an effective method without prior assumptions and prior processing, which may include denoising, outlier removal, normal estimation, data completion, or mesh reconstruction. It adapted *L1*-medians locally to a point set representing a 3D shape, giving rise to a one-dimensional structure, which was seen as the localized center of the shape. Thus, this would be an appropriate skeleton algorithm for use on the reconstructed point cloud, which might be uneven and incomplete. Although competitive skeletal results can be derived, time efficiency is seriously affected by the data size of the input point cloud. During SfM-MVS reconstruction processing, many redundant points were produced, which would lead to low efficiency for phenotypic trait estimation. Therefore, the original point cloud was downsampled through an octree-based methodology to balance the trade-off between efficiency and performance. The downsampled point cloud was set as the input for the skeletonization process. Diagrams showing the original point cloud and the generated skeleton are provided in Figure 1e,f.

### 2.6. Structural Phenotypic Trait Estimation

Structural phenotypic traits represent the morphological characteristics of tomato plants, such as plant volume, plant height, canopy height, canopy area, node number, and internode length, and the definitions and abbreviations of the phenotypic traits are illustrated in Table 1.

*Plant height:* Plant height is defined as the vertical distance from the ground to the highest point of the plant in its natural state, which can be represented by the height of the minimum bounding box (MMB), $H_B$.

*Plant volume:* Plant volume can be represented by the volume of the convex hull and the minimum bounding box (MMB). The convex hull is the smallest convex set that

contains all the points, while the MMB is defined as the minimum box that contains all the points of the tomato plant. Figure 1h shows the 3D tomato plant model enclosed by the MMB and convex hull. In this paper, we choose the convex hull volume to represent the plant volume.

*Stem length:* Stem length is defined as the sum of all internode lengths.

*Canopy area:* The canopy area is defined as the multiplication of the plant width and plant depth.

*Node number and internode length:* The main stem of the tomato plant is not typically straight, which will make the measurement of plant height difficult. A series of nodes are located around the main stem. The internode length is defined as the Euclidean distance between two adjacent nodes. The length together with the number of internodes determines the length of the main stem. Hence, an alternative strategy for stem length calculation is adopted in the study. The skeleton of the stalk is already generated on the point cloud, and it depicts the morphological structure. The nodes located in the main stem can be found by using a suitable search strategy. In this way, the internode length, representing the distance between two nodes along the main stem, can be calculated, and the stem length can be determined by summing all the internode lengths.

**Table 1.** Seventeen 3D phenotypic traits measured by the 3DPhenoMVS pipeline.

| Trait | Abbreviation |
| --- | --- |
| Plant height: height of the minimum bounding box | PH |
| Plant width: width of the minimum bounding box | PW |
| Plant depth: depth of the minimum bounding box | PD |
| Stem length: sum of all internode lengths | SL |
| Canopy area: product of plant width and depth | CA |
| Primary ratio: ratio of plant height to plant width | RP |
| Secondary ratio: ratio of plant height to plant depth | RS |
| Internode length: Euclidean distance between two adjacent nodes | IL |
| Node number | NN |
| Leaf number | LN |
| Leaf length: sum of all the Euclidean distances between two ends | LL |
| Leaf width: width of the minimum bounding box of the leaf | LW |
| Fruit volume: convex hull volume | FV |
| Fruit area: convex hull area | FA |
| Vertical diameter: maximum diameter on X-axis | VD |
| Transverse diameter: maximum diameter in YOZ plane | TD |
| Fruit shape index: ratio of vertical diameter to transverse diameter | FSI |

*Number of leaves, leaf length and leaf area:* The number of leaves denotes the total number of leaves in the plant, and it indicates the tomato plant's physiological age; thus, it is also involved in phenotypic trait estimation. In the early growth period of the tomato plant, it is identical to the number of nodes; however, it is different in the late growth period due to the pruning of old leaves. Leaf length measures the length of the leaf in 3D space, and it is estimated by summing all the Euclidean distances between two leaf ends. One end is the starting point of the petiole located at the main stem, and the other end is located at the end of the leaf. Leaf area is the most closely related and variable factor of yield. It measures the surface area of leaves in 3D space. To estimate one leaf area accurately, a mesh model for the 3D points representing this leaf is generated, and the sum of the area of all the meshes is the leaf area.

*2.7. Fruit Phenotypic Trait Estimation*

Fruit phenotypic traits represent the morphological characteristics of tomato fruits, such as transverse diameter, vertical diameter, the fruit shape index, fruit volume, and fruit

area, as shown in Figure 1l. The definitions and abbreviations of the fruit phenotypic traits are also illustrated in Table 1.

*Fruit volume:* Fruit volume can be represented by the volume of the convex hull of the generated fruit 3D point clouds.

*Fruit surface area:* Fruit surface area can be represented by the sum area of the surface meshes.

*Vertical diameter:* It is defined as the distance between the pedicle and the other end of the tomato fruit.

*Transverse diameter:* The diameter of the largest transverse section perpendicular to the vertical diameter.

*Fruit shape index:* Fruit shape index is defined as the rate between the vertical and transverse diameter.

### *2.8. 3D Point Cloud from Active Sensors*

To evaluate the performance of the reconstructed point clouds from images, LiDAR and structured light scanner were used to collect point clouds as reference data for comparison. As to the LiDAR system, a FARO Laser Scanner (FARO Focus S70, FARO Corporation, Lake Mary, Florida, America), following the indirect time of flight principle, was used for data collection in the greenhouse. A group of 3 target balls with a diameter of 0.15 m were placed in the scenario, and the sensor was mounted on a tripod at a consistent height. A total of 6 LiDAR frames covering the area of 10 m × 1.5 m were collected. Two LiDAR frames were registered and aligned together first with the aid of the target balls to produce one set of point cloud consisting of the first two LiDAR frames. Later, the other LiDAR frames were registered and aligned with the produced point clouds following the same procedures one by one. All the procedures were implemented by using the commercial software packages SCENE provided by FARO. A complete point cloud was generated, with an average error of 3.4 mm, which depicted the distance between the points from two LiDAR scans.

In regard to the structured light sensor, a 3D structured light scanner (Reeyee Pro, Wiiboox Corporation, Nanjing, China), leveraging white light into accurate measurements, was adopted for data acquisition of individual plant or tomato fruit in the indoor scenario. One tomato plant or fruit was placed on a rotary table with fluorescent targets controlled by a step motor. The rotation angle was set at 20°, and the commercial software Reeyee-Pro-v2 managed the data collection and point cloud generation in an automatic fashion. A complete point cloud of individual plant was created with error around 0.15 mm, and tomato fruit with error around 0.12 mm.

Based on the discussion above, both the accuracy of the point clouds generated by a LiDAR sensor and a structured light sensor were quite high, and thus, they were used as reference data for later accuracy evaluation.

### *2.9. Evaluation Indexes of Phenotypic Trait Extraction*

To evaluate the accuracy of the reconstructed point cloud in our pipeline, we used the point clouds created by a LiDAR scanner and structured light scanner as the reference data, and the Hausdorff distance between the two sets of point clouds were calculated [37]. The Hausdorff distance, which described the distance between proper subsets in the metric space, was defined as Equation (1).

$$H(A, B) = \max(h(A, B), h(B, A)) \tag{1}$$

$$h(A, B) = \max_{a \in A} \left\{ \min_{b \in B} \| a - b \| \right\} \tag{2}$$

$$h(B, A) = \max_{b \in B} \left\{ \min_{a \in A} \| b - a \| \right\} \tag{3}$$

where A and B denotes two sets A = {a$_1$, a$_2$, $\cdots$, a$_n$} and B = {b$_1$, b$_2$, $\cdots$, b$_m$}, h(A, B) is called the one-way Hausdorff distance from set A to set B, and vice versa.

Compared with manual measurement, R$^2$ and MAPE were used to evaluate the performance of the phenotypic trait extraction. The MAPE of the automatic measurement versus manual measurement was defined in Equation (4).

$$MAPE = \frac{1}{N} \sum_{i=1}^{N} \left| \frac{t_m - t_c}{t_m} \right| \times 100\% \tag{4}$$

In Equation (4), $N$ denotes the number of times the summation iteration occurs, $t_m$ denotes the measurement value, and $t_c$ denotes the value calculated from the 3D models.

### 3. Results

*3.1. 3D Reconstructed Point Cloud of Tomato Plants Covering Different Life Cycle*

For each individual tomato plant, one group of images was collected during the earlier growth stage, while two sets of images, for the upper part and lower part, were acquired once occlusion occurred during the later growth stage. Alignment was implemented to produce a complete point cloud with respect to the whole plant, as shown in Figure 3a–c. Time series 3D models of tomato plants covering the full growth period were created with high-quality and rich color information, and the structural changes of tomato plants during the different growth and development stages were also visualized, as shown in Figure 3d. After removing the leaf points manually, only the stalk points were retained, and the skeletonization algorithm was implemented on the downsampled stalk points to output skeletonized stalks with nodes. Figure 3e shows a side view of the skeletonized results. Node detection was conducted in an automatic way for node number and internode length calculation. The source code and detailed implementation are described in Supplementary Note S1.

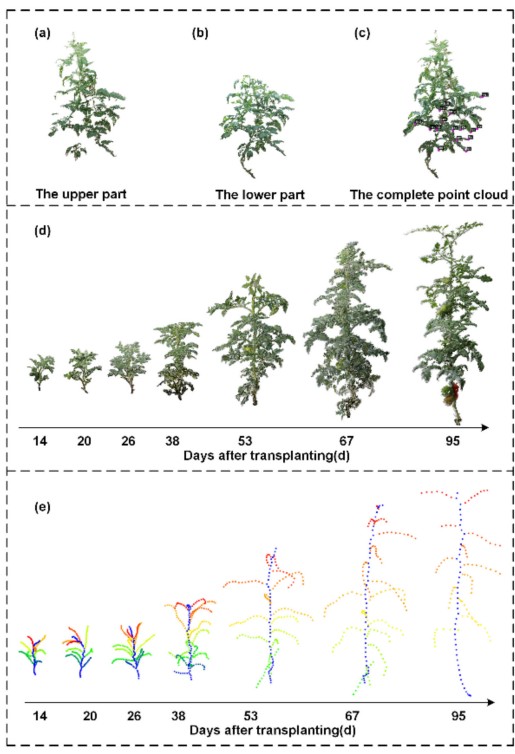

**Figure 3.** Visualization of reconstruction and stalk skeletonization. Point clouds for (**a**) the upper part and (**b**) lower part of one tomato plant and (**c**) a complete tomato plant model after alignment. (**d**) All growth stages of one tomato plant at 7 time points. (**e**) The stalk skeletonization results of one tomato plant at 7 time points.

### 3.2. Comparison of the Generated 3D Point Cloud from Images and Active Sensors

To evaluate the reconstruction result, active sensors, LiDAR and structured light scanner, were used to collect point clouds as reference data due to the high accuracy of them. The data collection was implemented at greenhouse scenario and individual plant level.

A FARO scanner (FARO Focus s70, FARO Corporation, Lake Mary, Florida, America) was utilized to scan the plants in the greenhouse. The results of generated point clouds and LiDAR based point clouds were visualized as shown regarding the tomato plants in Figure 4a–d. The LiDAR based point cloud of the tomato plants were segmented manually from the original point clouds as shown in Figure 4b, and the reconstructed point clouds were shown as Figure 4d. A closer look at the figures showed that the visualization effects of the reconstructed point cloud were compatible compared to the LiDAR-based point cloud, and the Hausdorff distance of the two sets of point clouds was below 0.93 cm.

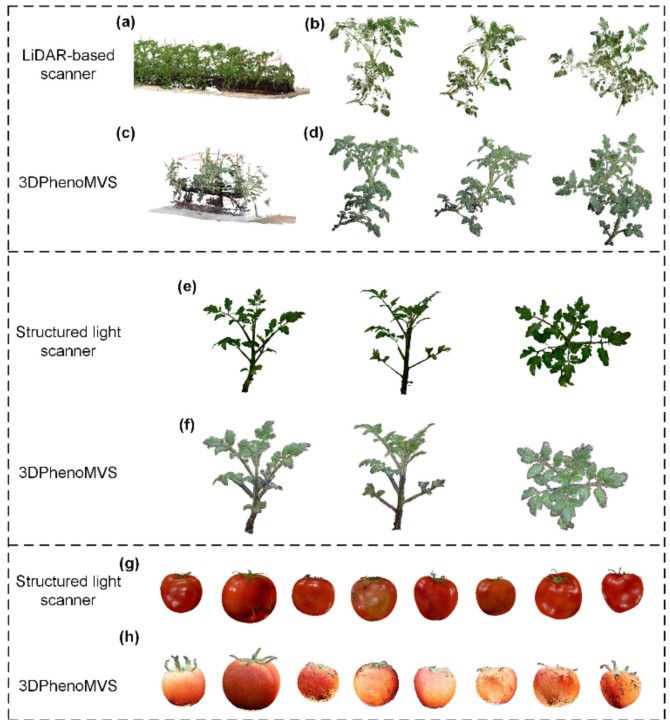

**Figure 4.** Comparison of point cloud acquired by Different Acquisition Methods. (**a**) Original point cloud acquired by laser scanning and (**b**) different perspectives of a tomato plant model. (**c**) Original point cloud acquired by 3D reconstruction and (**d**) different perspectives of a tomato plant model. Based on structured light imaging: (**e**) different perspectives of one tomato plant acquired by structured light sensor and (**f**) different perspectives of one tomato plant acquired by 3DPhenoMVS. (**g**) Tomato fruit models acquired by structured light imaging and (**h**) tomato fruit models acquired by 3D reconstruction.

In addition, more comparison experiments were implemented on different plants including the seedling state of tomato plant at individual plant level, and also tomato fruit, in order to demonstrate the effectiveness of the proposed pipeline, as shown in Figure 4e–h. The reconstructed point clouds were compared to the reference data, which were generated by a structured light scanner (Reeyee Pro 3D, Wiiboox Corporation, Nanjing, China). The Hausdorff distance between the two plant point clouds was below 0.74 cm, and the Hausdorff distances regarding tomato fruits were 0.46 cm, 0.55 cm, 0.74 cm, 0.66 cm, 0.51 cm, 0.42 cm, 0.98 cm, and 0.56 cm, respectively. Although some points were missing in a small area regarding the reconstructed point clouds, the characteristics of low-cost and

convenience made the proposed pipeline superior to the expensive LiDAR and structured light sensors.

### 3.3. Accuracy Evaluation on Phenotypic Traits

Manual measurements of phenotypic traits were also conducted on tomato plants to demonstrate the accuracy of the algorithm proposed in this research. Since true values of 6 phenotypic traits including plant height, stem length, internode length, transverse diameter, vertical diameter, and node number can be measured manually and counted, the accuracy estimation was conducted on them, and the comparative results are shown in Figure 5a–f. The results showed that the $R^2$ values of plant height, stem length, internode length, leaf number, transverse diameter, and vertical diameter of tomato plants were 0.97, 0.86, 0.95, 0.76, 0.93 and 0.72, respectively. The MAPE of the above-mentioned six parameters were 17.23%, 13.89%, 20.62%, 8.20%, 2.69%, and 14.19%, respectively.

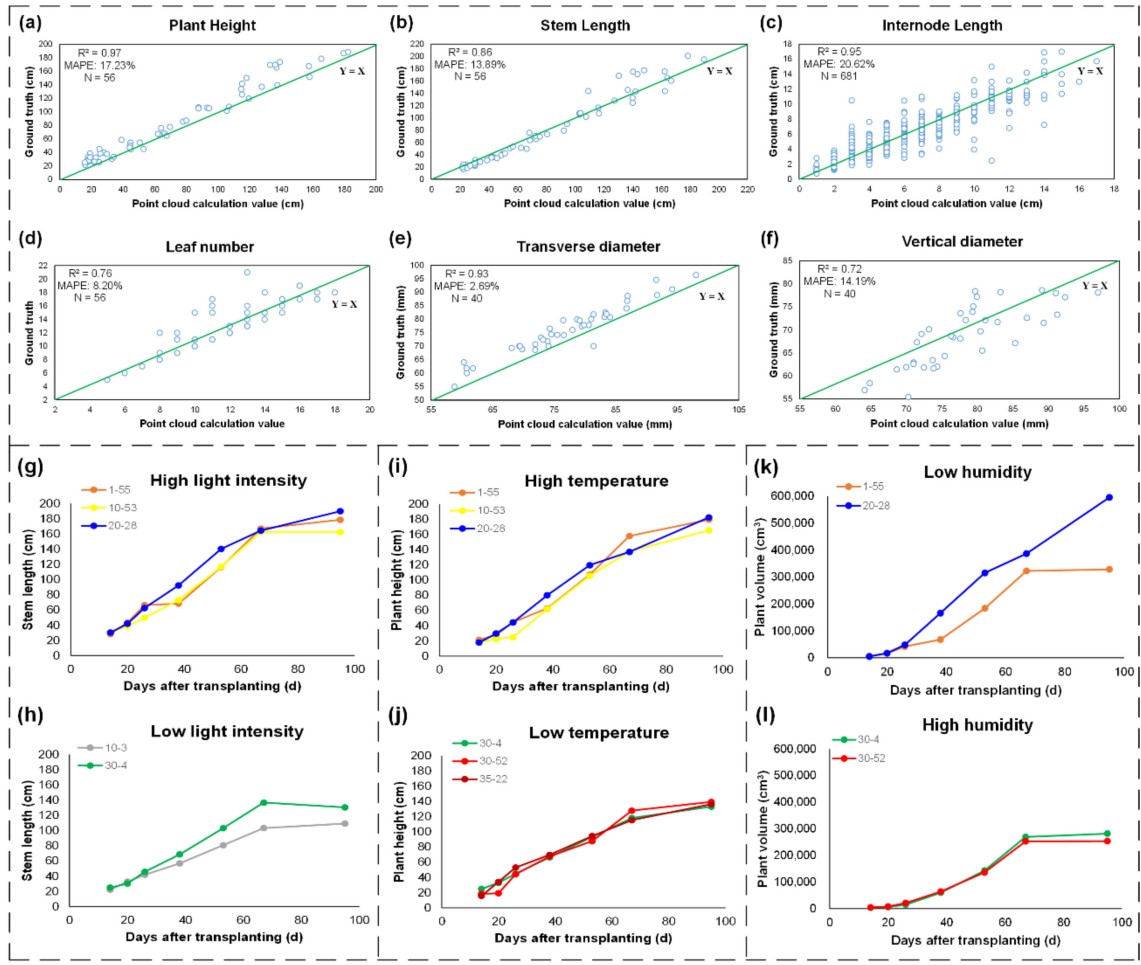

**Figure 5.** The performance evaluation of 3D trait extraction and growth variation with different environments. (**a**) Plant height, (**b**) stem length, (**c**) internode length, (**d**) leaf number, (**e**) transverse diameter, and (**f**) vertical diameter. (**g**,**h**) Influence of light intensity variation on stem length, (**i**,**j**) influence of temperature variation on plant height, (**k**,**l**) influence of humidity variation on plant volume.

### 3.4. Growth Variation with Different Environments

To investigate the environmental influence on the growth of tomato plants, the growth variation in different environments was measured and validated. We divided the locations of each environmental parameter into two categories according to the environmental difference distribution map (Figure 2b), such as locations with high intensity and low intensity,

locations with high humidity and low humidity, and locations with high temperature and low temperature. The tomato samples were classified into two categories by their locations. All the phenotypic traits were analyzed in combination with the environmental parameters. Among them, stem length, plant height, and plant volume show great difference as the environmental parameters varied, and the other parameters did not present such trends. Thus, we used three of them to analyze the influence of the environmental differences to the growth of tomato plants. As shown in Figure 5g–l, the three environmental factors showed various influences on the growth of the tomato plants in the greenhouse. Tomato plants located in areas with higher intensity and moderate temperature and humidity show longer stem lengths than plants with lower light intensity, such as plants 1-55, 10-53, and 20-28. Plants 10-3 and 30-4, located at the other end of the greenhouse, show shorter stem lengths and lower growth rates of stem length. Considering the influence of temperature, the tomato plants (plants 1-55, 10-53, and 20-28) in the higher-temperature area show relatively large plant height, while plants 30-4, 30-52, and 35-22 show lower plant height. Two rows of plants (rows 30 and 35) were located at the west end of the greenhouse, where a reservoir and wet curtains were equipped; hence, the temperature was lower than that on the other side. Regarding humidity, the two plants located in the 30th row with high humidity, plants 30-4 and 30-52, showed lower plant volumes, while the plants located in the area with lower humidity showed higher volumes, as shown in Figure 5k,l.

In addition, tomato fruit weight for each individual plant was measured to further investigate the environmental influence. As shown in Figure 2b, higher yields occurred in plants 1-55, 10-53, and 20-28, while plants 10-3 and 30-4 showed lower yields. Thus, by testing the eight samples in the greenhouse, it was concluded that higher yield can be expected for plants with higher light intensity and moderate temperature and humidity.

*3.5. Measuring Efficiency Evaluation*

In this study, a consumer digital camera was used to collect image data for each individual tomato plant. While the plant was in the seedling stage, approximately 100 images were taken, and this took approximately 15 min; in the later stage, two sets of images were acquired, and approximately half an hour was needed. These procedures were all implemented on a Windows 10 operating system, and the central processing unit (CPU) was an Intel Core i7-8700 (16 GB of random-access memory). SfM was implemented by VisualSFM, and approximately 60 min were needed to generate one point cloud for one tomato plant. Plant volume was represented by the convex hull of the generated 3D points, and approximately 1 min was needed to calculate the volume of a single plant at 7 time points. Plant height, plant width, and plant depth were represented by the minimum bounding box of the generated 3D points, and determining these values took approximately 20 s. Leaf points were separated from stalk points by minor manual work, which took approximately 2 min. After that, the *L1*-median skeletonization algorithm was carried out on the downsampled stalk points, and it took approximately 1–2 min to process 10k points. Node detection was conducted on the skeletonized stalk points to calculate internode length automatically (the source code for node detection can be downloaded as explained in Supplementary Note S1). Thus, the total time consumption for extracting all the morphological traits for one tomato plant was 80 min.

Among all the procedures, the 3D reconstruction process took most of the time, which is mainly due to the structural complexity of tomato plants, as can be seen from the visualization results. Higher reconstruction efficiency would be achieved with a more advanced computer configuration. In addition, the proposed pipeline has another significant advantage: all the node detection and internode length calculation procedures regarding the stalk of one individual tomato plant at different time points are implemented in a fully automated way. The phenotypic trait calculation regarding leaves follows the same pipeline. In other words, it is not necessary to build the correspondences for certain organs, such as nodes and leaves, at different growth stages in order to investigate the temporal trait changes regarding one plant organ.

### 3.6. The Generalization Ability of the Proposed 3D Phenotyping Pipeline

To demonstrate the generalization ability of the proposed 3D phenotyping pipeline, six other plant species, including maize, cotton, rapeseed, tobacco, chili, and eggplant, were chosen as the plant samples. The 3D reconstruction and point cloud skeletonization procedures were implemented, and the results shown as Figure 6 demonstrated the potential of the proposed pipeline on other plants. Additionally, the 3D phenotypic traits can be further estimated on the generated point clouds and the skeletonized results automatically.

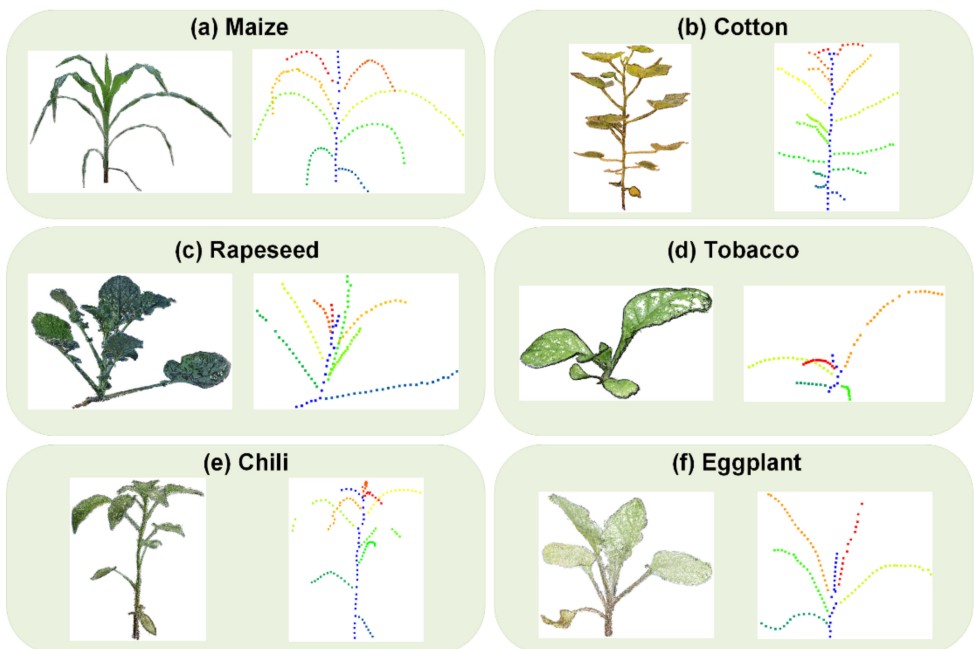

**Figure 6.** Visualization of reconstructed point clouds and stalk skeletonization results regarding 6 species of plants.

## 4. Discussion

### 4.1. Comparison of 3D Data Collection Methods

In this study, three representative 3D data acquisition methods were used to generate 3D point cloud, including LiDAR-based scanner, structured light scanner, and 3D reconstruction from images by considering the accuracy of point cloud, post-processing difficulties, cost, data collection efficiency, and complexity of data collection scenario. As shown in Figure 4a–h, the accuracy of point clouds generated by LiDAR or structured light sensor is quite high [9], and the local density of the point cloud is more uniform, which reduces the difficulty of post-processing, and enables high-throughput data collection. But the high price of the sensor has limited its wide application (LiDAR: $78,500; structured light sensor: $20,000; digital camera: $754). Compared to the reconstructed point cloud from images, structured light sensor shows comparable visualization effect, higher degree of automation, and higher cost. However, the plants need to be placed on a specific rotary table, which has certain limitations on the size of the plants and the data collection must be completed in an indoor scenario. In comparison to the LiDAR and structured light systems, 3D reconstruction from images not only achieved good point cloud quality for tomato plants, but also show dominant superiorities, such as low cost, high flexibility, and easily accessible data collection.

Regarding the tomato fruits, the reconstructed point clouds of tomato fruits were compared with the points generated by using structured light sensors. As shown in Figure 4, the visualization result of structured light sensor is better in terms of color and texture. Due to the smoothness of tomato fruit, highlight effects (specular reflection) happened during the data collection using structured light sensor, and thus, it is necessary

to use markers as auxiliary apparatus, which increases the complexity of the data collection. Both the point clouds generated by structured light sensor and images show similar effects regarding the morphological and structural information of tomato fruits.

*4.2. Reconstruction Strategy*

To date, the development of 3D reconstruction technology based on multiple images has promoted a wide range of applications for this technology in 3D plant phenotyping because of its advantages as a non-contact, low-cost, and high-precision method. Fixed cameras and moving cameras are the two main scenarios used for plant reconstruction. In the fixed-camera scenario, the advantage of this approach is its high time efficiency for 3D reconstruction, and fewer images are required as input data in this case than in the moving-camera scenario. However, some notable types of defects cannot be prevented by this solution. The prerequisite for fixed-camera reconstruction is to calculate the accurate position and orientation parameters of all cameras, i.e., the intrinsic and extrinsic parameters of the cameras, which are determined through a calibration algorithm. This increases the complexity of the approach, especially for researchers who do not have computer vision experience and backgrounds. Both the correspondence-based and carving-based reconstruction methods are involved in the calibration procedures; thus, other accessibility settings, such as a checkerboard [15] or a cube with checkerboard patterns [21], were used for the calibration procedure. In addition, multiple image data collection was completed with the assistance of a turntable or stand [12], an electronic rotary [38], and other auxiliary facilities. Meanwhile, the experiment was limited to specific indoor scenes because of the utilization of fixed cameras and other accessibility settings.

Regarding the investigation of the environmental influence on tomato plants, all the environmental conditions should remain the same, and the tomato plants were fixed at the same locations during all growth stages. Hence, moving-camera solutions were more applicable in this study. In contrast to the fixed-camera scenario, complicated calibration procedures were avoided, and the tomato plants could stay in their original environmental conditions without human intervention.

Previous studies successfully applied 3D reconstruction algorithms, such as Poisson surface reconstruction [39], and $\alpha$-shape based algorithm [25,40]. These algorithms mainly focused on the surface reconstruction of leaves and canopies and output a meshed surface model. The estimation of phenotypic traits, such as leaf area, was conducted directly on the surface mesh models. However, this procedure was not applicable for node counting, internode length estimation. Hence, we use the reconstructed point cloud for the later procedures rather than surface mesh model.

*4.3. The Environmental Influences on the Growth and Yield*

It has been demonstrated that environmental differences influence the growth of tomato plants. No significant growth difference regarding the total plant area, plant volume, stem length, or canopy area was shown in the early periods, while a great difference was shown in the late growth stages, especially during the fruit growth periods.

Differences in light intensity, temperature, and humidity can lead to differences in plant growth. In the northern part of the greenhouse, the light intensity was weak due to the shelter. Wet curtains led to higher humidity, and draught fans and vents led to lower temperatures. In the same row, the difference in temperature and humidity was very small. Interestingly, the plant growth on the south side was better than the plant growth on the north side. For example, the overall growth of plants 1-55, 10-53, and 30-52 was better than that of plants 1-1, 10-3, and 30-4, respectively. This may indicate that light intensity has a greater impact on plant growth. It should be noted that the three environmental parameters in the central area were not the highest or lowest, but the yield of plant 20-28 was the highest. The reason was mainly because the environmental changes in the central area were not as drastic as those on the east or west sides when performing normal switching operations.

To investigate the environmental influence on the yield, the fruit number and fruit weight were also measured manually at the mature stage. As shown in Figure 2b, the yields of plants 1-55, 10-53, and 20-28 were higher than those of the other 5 plants, indicating that higher light intensity with moderate temperature and humidity will contribute to better yield. In other words, slight differences in environmental changes, especially in light intensity, caused significant differences in individual plant growth and ultimately affected the yields of tomato plants.

*4.4. Outlook and Perspectives*

The application of the proposed low-cost and open-access pipeline has been demonstrated on tomato plants covering all growth stages. At the same time, the pipeline has been applied to other plants, such as cotton, corn, rapeseed, chili, eggplant, and tobacco. To further exploit the application of the proposed 3DPhenoMVS, two aspects need to be considered in future research: (1) The pipelines could be applied to plant roots, in order to generate a whole plant model combining the above-ground and under-ground parts. (2) To exploit the relations between plant growth and environmental conditions, further experiments could be designed, for example, using a controlled light source to monitor the plant growth, such as leaf color or the rate of photosynthesis. Except the image based low-cost phenotyping pipeline, the applications of new emerged low-cost LiDAR systems in phenotyping needs to be further explored.

**5. Conclusions**

In this work, a low-cost and open-access temporal 3D phenotyping pipeline, 3DPhenoMVS, was proposed. Based on the analysis of environmental differences in a greenhouse, eight tomato plants were chosen to generate 3D point clouds. The 3D reconstructed models could show the morphological structure covering the whole life cycle. Skeletons of stalks were also produced on the point clouds to perform accurate phenotypic trait calculations. Seventeen phenotypic traits were calculated, and the $R^2$ values regarding stem length, plant height, internode length and transverse diameter were more than 0.85. In addition, slight environmental differences in the greenhouse had different influences on tomato plant growth, and the yield difference also demonstrated this. Specifically, stronger light intensity with moderate humidity and temperature contributed to higher yield. In addition, generalization potential of the proposed pipeline was also tested and demonstrated on other species of plants. Overall, the proposed low-cost and open-access phenotyping pipeline could benefit breeding, cultivation research, and functional genomics in the future.

**Supplementary Materials:** The following supporting information can be downloaded at: https://www.mdpi.com/article/10.3390/agronomy12081865/s1. Note S1: The instruction of 3DPhenoMVS; Video S1: Operating procedure for 3DPhenoMVS [33–36,41]; Video S2: The whole grow stages of plant 1-1 at 7 time points; Video S3: The whole grow stages of plant 1-55 at 7 time points; Video S4: The whole grow stages of plant 10-3 at 7 time points; Video S5: The whole grow stages of plant 10-53 at 7 time points; Video S6: The whole grow stages of plant 20-28 at 7 time points; Video S7: The whole grow stages of plant 30-4 at 7 time points; Video S8: The whole grow stages of plant 30-52 at 7 time points; Video S9: The whole grow stages of plant 35-22 at 7 time points.

**Author Contributions:** Data curation, R.Z. and Y.W.; formal analysis, Y.W.; software, S.H. and H.R.; writing—original draft preparation, R.Z. and Y.W.; writing—review and editing, W.Y. and R.Z.; visualization, Y.W.; supervision, W.Y. and R.Z. All authors have read and agreed to the published version of the manuscript.

**Funding:** This work was supported by grants from the National Natural Science Foundation of China (U21A20205), Key projects of Natural Science Foundation of Hubei Province (2021CFA059), Fundamental Research Funds for the Central Universities (2021ZKPY006, 2662022JC004), and HZAU-AGIS Cooperation Fund (SZYJY2022014).

**Data Availability Statement:** The user guidelines can be downloaded in Supplementary Materials. All the phenotypic data and images can be viewed and downloaded via the links https://drive.

google.com/drive/folders/10bx0-8j9ABWKTY07WNiBtwQWuB5LHqgZ?usp=sharing (accessed on 29 June 2022) and http://plantphenomics.hzau.edu.cn/download_checkiflogin_en.action (accessed on 29 June 2022).

**Acknowledgments:** We thanked Harvest-Code Technology (Nanjing) Ltd. provided the materials and experimental resources.

**Conflicts of Interest:** The authors declare no conflict of interest.

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
