# Peer review of "3DPhenoMVS: A Low-Cost 3D Tomato Phenotyping Pipeline Using 3D Reconstruction Point Cloud Based on Multiview Images"

_agronomy, doi:10.3390/agronomy12081865_

Round 1

Reviewer 1 Report

The authors developed a 3D phenotyping pipeline of Tomato plants using multiview images and 3D reconstruction point clouds. Overall, the manuscript is in good shape; however, I suggest the authors to address few minor changes. If the changes are addressed, I recommend this manuscript to be published in the journal.

·       Line 15: Please include the range of R-squared. For instance, the R-squared ranged from 0.xx to 0.yy.

·       Line 62: I suggest the authors to explain briefly what “binocular stereo vision” is.

·       Line 79: Consider replacing the work “papers” with “literature”.

·       Line 114: Figure 1a is hard to understand. The environmental conditions should be clearly mentioned here.

·       In Section 3.3, the accuracy of phenotypic traits have been mentioned. However, I can only see discussion about 6 traits. I suggest the authors to include the results of all 19 traits here in Figure 5 or may be create a table with evaluation metrics of all 19 traits.

·       Since there are some good concluding remarks about the experiment, the authors should include a Conclusion section at the end. I believe these points have been discussed in Section 4.4 already. Therefore, the authors can simply use this section as the Conclusion. This will be good for the readers who want to know a quick summary result of the whole experiment.

Author Response

We thank the reviewers and editor for their constructive comments and suggestions of this manuscript. We addressed each point in turn below. The answers and revisions to each point have been highlighted using the blue and red fonts to make it easily visible to the editor and reviewers. Please see the attachment. Should there been any other suggestions and comments, please feel free to contact us.

Response to reviewer#1

Thank you very much! You have greatly improved our MS. We highly appreciate your suggestions. We have carefully revised our MS and incorporated your revisions.

Comment 1: Line 15: Please include the range of R-squared. For instance, the R-squared ranged from 0.xx to 0.yy.

Response: Thank you very much for pointing out this. The R-squared has been modified as suggested. The modified version is as follows.

 “The results showed that the R2 values between the phenotypic traits and the manual measurements were in the range 0.72 to 0.97.

Comment 2: Line 62: I suggest the authors to explain briefly what “binocular stereo vision” is.

Response: Thank you very much for pointing out this. We have complemented the explanation of “binocular stereo vision”. The modified version is as follows.

“Compared to binocular stereo vision, which is a method to derive 3D information based solely on the relative positions of the object in the two cameras, multiview stereo technology is more applicable because a complete point cloud of an object can be generated [11]

Comment 3: Line 79: Consider replacing the work “papers” with “literature”.

Response: Thank you very much for pointing out this. The “papers” has been replaced with “literature”.

Comment 4: Line 114: Figure 1a is hard to understand. The environmental conditions should be clearly mentioned here.

Response: Thank you very much for pointing out this.  Figure 1 depicts the whole pipeline, and Figure 1a demonstrates the first step, environmental monitoring, which is used for tomato plant selection as research target samples based on the environmental differences. To avoid misunderstanding, we change the subheading of Section 2.1 as “Pipleline of 3D PhenoMVS”, and the subheading of Section 2.2 as “Environmental conditions of plant material and image collection”. The detailed description of environmental monitoring is given in Section 2.2 from line 161 to line 171.

Comment 5: In Section 3.3, the accuracy of phenotypic traits have been mentioned. However, I can only see discussion about 6 traits. I suggest the authors to include the results of all 19 traits here in Figure 5 or may be create a table with evaluation metrics of all 19 traits.

Response: Thank you very much for pointing out this. We agreed with the reviewer that we only discuss the accuracy of 6 traits. The reasons are as follows.

  • Among all the 19 traits estimated, true values for plant volume would be acquired by water displacement method, and accurate leaf area would be measured by leaf area meter. Both the methods were destructive and would cause destruction to tomato plants, which is not consistent with our research object, since we are focusing on the whole plant growth stages of tomato plants.
  • The calculation of leaf length and width is consistent with the internode length and plant height, thus, the accuracy would be consistent with the estimation of internode length and plant height.
  • Regarding the tomato fruit volume and area, we were focusing on the whole growth stages of the tomato plant, and estimation of tomato fruit traits were used as complementary materials to demonstrate the effectiveness of the proposed pipeline, thus, the accuracy assessment were not analyzed in depth in the study.
  • The remaining other traits, such as primary ratio, secondary ratio, and fruit shape index, were derived from the 6 traits, including plant height, width, depth, and others, therefore, no precision assessment was performed on the them.

Comment 6: Since there are some good concluding remarks about the experiment, the authors should include a Conclusion section at the end. I believe these points have been discussed in Section 4.4 already. Therefore, the authors can simply use this section as the Conclusion. This will be good for the readers who want to know a quick summary result of the whole experiment.

Response: Thank you very much for pointing out this, which helps a lot to improve the manuscript.  We have removed the conclusion words in Section 4.4 and included a Conclusion section at the end The modified version is as follows.

“5. Conclusion

In this work, a low-cost and open-access temporal 3D phenotyping pipeline, 3DPhenoMVS, was proposed. Based on the analysis of environmental differences in a greenhouse, eight tomato plants were chosen to generate 3D point clouds. The 3D reconstructed models could show the morphological structure covering the whole life cycle. Skeletons of stalks were also produced on the point clouds to perform accurate phenotypic trait calculations. Nineteen phenotypic traits were calculated, and the R2 values regarding stem length, plant height, internode length and transverse diameter were more than 0.85. In addition, slight environmental differences in the greenhouse had different influences on tomato plant growth, and the yield difference also demonstrated this. Specifically, stronger light intensity with moderate humidity and temperature contributed to higher yield. In addition, generalization potential of the proposed pipeline was also tested and demonstrated on other species of plants. Overall, the proposed low-cost and open-access phenotyping pipeline could benefit breeding, cultivation research, and functional genomics in the future.”

 Again, thanks a lot for your positive comments and suggestions, we really appreciate that.

Reviewer 2 Report

In this paper. The authors proposed a low-cost and open-access 3D phenotyping pipeline with the use of structures from motion algorithms, named 3DPhenoMVS, to calculate and analyze phenotypic traits of tomato plants during the whole life cycle. The results showed good correlations between the 14 phenotypic traits and the manual measurements. Also, the results showed that stronger light intensity and moderate temperature and humidity contribute to higher biomass and higher yield. The pipeline could be applied to other plant species, which demonstrated the proposed pipeline will benefit plant breeding, cultivation research, and functional genomics in the future. The paper is well prepared and worth publishing in Agronomy.

Author Response

Thank you very much for your positive comments and valuable suggestions on our manuscript, which is very encouraging, and we really appreciate that.

Reviewer 3 Report

The manuscript described an RGB photogrammetry-based tomato phenotyping study. Despite the apparent effort that the authors have made towards experiment conduction, data analysis and manuscript writing, in my opinion, there is a fundamental issue with the study that unfortunately leads to my rejection decision: the ambitious scope of the study is not fully examined in the experiment. The study added a series of data processing algorithms after point cloud reconstruction to predict 19 traits of tomato. However, I do not see a comprehensive, careful evaluation of the pipeline for every single tomato trait that the authors claimed to be predictable by the pipeline, which is a must to support the conclusions. I would like to see figures such as Figure 5a-f for all 19 traits, and I hope because of my rejection decision the authors can have ample time conducting necessary additional experiment and data analysis to truly enhance the quality of the manuscript before resubmission. Below are my various observations. Some are major comments that need to be thoroughly addressed.

·       Line 33, weak logic, I don’t see how it’s urgent based on the argument.

·       Line 55, the statement may be true 3 years ago. Low-cost lidars are available now.

·       Line 98, 101-104, past tense should be used.

·       Line 117 118, 193-194 why two point clouds? Two sets of images could be used for reconstructing one point cloud.

·       Line 196-200, how many images were captured for each fruit?

·       Section 2.4, add a graph for illustration.

·       Section 3.2 messy structure. Most of the content should be in the method and material section. Lidar and structured light scanner data collection methods are not described in detail at all. What are the models of the sensors?

·       Section 3.2, comparing point clouds without using any quantitative metrics is meaningless.

·       Abrupt and strange to have eggplant and chili in section 3.2

·       Section 3.3, again messy structure and much of the content should be in the method section.

·       Figure 5a-f should be separated from Figure 5g-l, and Figure 5g-l should be in section 3.4.

·       Figure 5g-l, are those average values? What about the rest traits regarding light intensity, temperature and humidity? Why some figures have three lines while others only have two? A consistent analysis is missing.

·       Section 3.6 is irrelevant to the study. Presenting more information does not necessarily improve manuscript quality.

·       Line 439, how do you know laser point cloud accuracy is the highest?

Author Response

We thank the reviewers and editor for their constructive comments and suggestions of this manuscript. We addressed each point in turn below. The answers and revisions to each point have been highlighted using the blue and red fonts to make it easily visible to the editor and reviewers. Should there been any other suggestions and comments, please feel free to contact us.

Response to reviewer#3

Thank you very much! You have greatly improved our MS. We highly appreciate your suggestions. We have carefully revised our MS and incorporated your revisions.

Comment 1: The study added a series of data processing algorithms after point cloud reconstruction to predict 19 traits of tomato. However, I do not see a comprehensive, careful evaluation of the pipeline for every single tomato trait that the authors claimed to be predictable by the pipeline, which is a must to support the conclusions. I would like to see figures such as Figure 5a-f for all 19 traits.

Response: Thank you very much for pointing out this. We agreed with the reviewer that we only discuss the accuracy of 6 traits. The reasons are as follows.

  • Among all the 19 traits estimated, true values for plant volume would be acquired by water displacement method, and accurate leaf area would be measured by leaf area meter. Both the methods were destructive and would cause destruction to tomato plants, which is not consistent with our research object, since we are focusing on the whole plant growth stages of tomato plants.
  • The calculation of leaf length and width is consistent with the internode length and plant height, thus, the accuracy would be consistent with the estimation of internode length and plant height.
  • Regarding the tomato fruit volume and area, we were focusing on the whole growth stages of the tomato plant, and estimation of tomato fruit traits were used as complementary materials to demonstrate the effectiveness of the proposed pipeline, thus, the accuracy assessment were not analyzed in depth in the study.
  • The remaining other traits, such as primary ratio, secondary ratio, and fruit shape index, were derived from the 6 traits, including plant height, width, depth, and others, therefore, no precision assessment was performed on the them.

Comment 2: Line 33, weak logic, I don’t see how it’s urgent based on the argument.

Response: Thank you very much for the good suggestions. We have modified it as follows based on your suggestion.

“It is essential to estimate the phenotypic traits of tomato plants and explore phenotypic trait variation during different growth stages, which can help to address and understand the relation between tomato plant growth and the surrounding environment. Although great advances have been made in tomato breeding, more efforts should be made to characterize objective, reliable and informative measurements of phenotypic traits to push breeding further [2]. Thus, sustainable breakthrough in tomato phenotyping is urgently needed.”

Comment 3: Line 55, the statement may be true 3 years ago. Low-cost lidars are available now.

Response: Thank you very much for pointing out this. We have investigated the new low-cost LiDARs emerged in the recent years from different aspects and the reasons why we don’t use LiDARs are as follows.

  • Regarding the dynamic LiDARs, including Velarray M1600, RIEGL VQ-840-G, HESAI Pandar128, etc., the prices are under 80,000 RMB in China. However, they cannot meet the demand of this study because they are mainly used for large/medium scale scenarios, such as unmanned driving and forestry, and the accuracy is not high, i.e., we could not acquire point clouds of leaves, stems and other organs with a qualified accuracy.
  • Regarding the close range LiDARs, such as HandyScan (CreaForm), MarvelScan (ZG), the prices are really high though the point clouds can be produced at high precision.
  • Other low-cost depth cameras are also investigated, such as Intel Realsense D435 and Kinect, however, the generated point clouds cannot meet the demand as shown in the following figure, which are the different perspectives of cotton plants. It can be seen from the figure that lots of points were missing and they can hardly be used for phenotypic trait estimation directly.

Figure. Three perspectives of cotton plant point clouds acquired by Intel Realsense D435

Comment 4: Line 98, 101-104, past tense should be used.

Response: Thanks for your nice suggestions. We have corrected these mistakes based on your suggestions. The modified version is as follows.

“Thus, 3DPhenoMVS, a low-cost and open-access 3D tomato phenotyping and growth analysis pipeline using a 3D reconstruction point cloud based on multiview images, was proposed to address the abovementioned issues. In addition, with 3D traits, we investigated the environmental influence on tomato plant growth and yield in greenhouses.”

Comment 5: Line 117 118, 193-194 why two point clouds? Two sets of images could be used for reconstructing one point cloud.

Response: Thank you very much for pointing out this. Point cloud registration is mainly used to reconstruct large sized tomato plant at the later stage of growth.  Theoretically, all the images can be input for reconstruction of the 3D plant models. In practice, the computer will crash once we use large number of images for reconstruction due to the limitations of the computation power. Hence, two point clouds were produced and registration and alignment were conducted to generate a complete plant model.

Comment 6: Line 196-200, how many images were captured for each fruit?

Response: Thank you very much for pointing out this. The number of images was about 100. We have supplemented the information in line 200-201.

“Generally, around 100 images were taken for one fruit.”

Comment 7: Section 2.4, add a graph for illustration.

Response: Thank you very much for pointing out this. Figures and detailed descriptions for fruit phenotypic trait estimation were in the supplementary material due to the limitations of the maximum number of figures. We have revised the description in line 235-237.

The user guidelines can be downloaded in Supplementary Materials. All the phenotypic data and images can be viewed and downloaded via the links:

https://drive.google.com/drive/folders/10bx0-8j9ABWKTY07WNiBtwQWuB5LHqgZ?usp=sharing  

and http://plantphenomics.hzau.edu.cn/download_checkiflogin_en.action.

 “The measurements of vertical diameter and transverse diameter are shown as Figure 1l, and the source code and detailed implementation are described in Supplementary Note 1.”

Comment 8: Section 3.2 messy structure. Most of the content should be in the method and material section. Lidar and structured light scanner data collection methods are not described in detail at all. What are the models of the sensors?

Response: Thank you very much for pointing out this. In section 3.2, we choose LiDAR and structured light scanner as the evaluation indicators to judge the quality of the reconstructed point cloud in this study, and we put emphasis on the generation of point clouds by using multiple images not the data acquisition methods. Thus, the data acquisition by using LiDAR and structured light sensor were not described in detail.

Regarding the models of the two sensors, please see lines 353 and 376, respectively. The detailed information is as follows.

“A FARO scanner (FARO Focus s70, FARO Corporation, America), was utilized to scan the plants in the greenhouse.”

The reconstructed point clouds were compared to the reference data, which were generated by a structured light scanner (Reeyee Pro 3D, Wiiboox Corporation, China).”

Comment 9: Section 3.2, comparing point clouds without using any quantitative metrics is meaningless.

Response: Thank you very much for the good suggestion, and we totally agree with it. Thus, we analyze the accuracy by comparing the generated point clouds to the reference data, namely, point clouds created by LiDAR or structure light sensor. Thus, Hausdorf distance was introduced in the manuscirpt, and the reference literature was also listed. The revised version for this part is as follows.

“To evaluate the accuracy of the reconstructed point cloud in our pipeline, we used the point clouds created by a LiDAR scanner and structured light scanner as the reference data, and the Hausdorf distance between the two sets of point clouds were calculated [37]. The Hausdorff distance described the distance between proper subsets in the metric space was defined as Equation 1.

(1)

(2)

(3)

Where, A and B denotes two sets  and , h(A, B) is called the one-way Hausdorff distance from set A to set B, and vice versa.”

Line 359:

“and the Hausdorf distance of the two sets of point clouds was below 0.93cm.”

Line 377-379:

“The Hausdorff distance between plant point clouds was below 0.74cm, and the Hausdorff distances regarding tomato fruits were 0.46cm, 0.55cm, 0.74cm, 0.66cm, 0.51cm, 0.42cm, 0.98cm, and 0.56cm, respectively.”

Comment 10: Abrupt and strange to have eggplant and chili in section 3.2.

Response: Thank you very much for the good suggestion, and we totally agree with it. Thus, we have removed the plant models of chili and eggplant in Section 3.2.

“Figure 4. Comparison of point cloud acquired by Different Acquisition Methods. (a) Original point cloud acquired by laser scanning and (b) different perspectives of a tomato plant model. (c) Original point cloud acquired by 3D reconstruction and (d) different perspectives of a tomato plant model. Based on structured light imaging: (e) different perspectives of one tomato plant acquired by structured light sensor and (f) different perspectives of one tomato plant acquired byPhenoMVS., (f) one chili plant and (g) one eggplant plant. Based on 3D reconstruction: (h) one tomato plant, (i) one chili plant and (j) one eggplant plant. (gk) Tomato fruit models acquired by structured light imaging and (h), (l) tomato fruit models acquired by 3D reconstruction.

In addition, more comparison experiments were implemented on different plants in-cluding the seedling state of tomato plant, chili plant, eggplant plant at individual plant level, and also tomato fruit, in order to demonstrate the effectiveness of the proposed pipe-line, as shown in Figure 4e-4hl. The reconstructed point clouds were compared to the ref-erence data, which were generated by a structured light scanner (Reeyee Pro 3D, Wiiboox Corporation, China). Although some points were missing in small area regarding the re-constructed point clouds, the characteristics of low-cost and convenience made the pro-posed pipeline more superior to the expensive LiDAR and structured light sensors.”

Comment 11: Section 3.3, again messy structure and much of the content should be in the method section.

Response: Thank you very much for the valuable suggestions. We have updated the description in section 2.8 (lines 322-327).

“2.8. Evaluation indexes of phenotypic trait extraction

Compared with manual measurement, R2 and MAPE were used to evaluate the performance of the phenotypic trait extraction. The MAPE of the automatic measurement versus manual measurement was defined in Equation 4.

                        (4)

In Eqs. 4, N denotes the number of times the summation iteration occurs, tm denotes the measurement value, and tc denotes the value calculated from the 3D models.”

Comment 12: Figure 5a-f should be separated from Figure 5g-l, and Figure 5g-l should be in section 3.4.

Response: Thank you very much for pointing out this. Mainly a typography issue, since the magazine limits the total number of figures, we put the figures in sections 3.3 and 3.4 together.

Comment 13: Figure 5g-l, are those average values? What about the rest traits regarding light intensity, temperature and humidity? Why some figures have three lines while others only have two? A consistent analysis is missing.

Response: Thank you very much for pointing out this. We divided the locations of each environmental parameter into two main categories through the environmental difference distribution map, such as locations with high intensity and low intensity, locations with high humidity and low humidity, and locations with high temperature and low temperature. The tomato samples were classified into different categories according to their locations. The tomato plants with obvious differences of phenotypic trait were analyzed by combining their environmental difference.

As shown in Figure 5g and 5h, plants 1-55, 10-53 and 20-28 were located at regions with high light intensity, and plants 10-3 and 30-4 were located at regions with low light intensity. By comparison, it was found that they had the most obvious difference in the stem length. The latter two environmental factors were also analyzed in the same way, respectively.

The lines of different colors in the figure represent different individual plants. Due to different environmental parameters, such as the location where the plant 10-53 grew (as shown in figure 2b), the light intensity, temperature, and humidity are high, so figure 5k does not have plant 10-53. Thus, some figures have three lines while others have two lines.

We put the consistent analysis in section 4.3 (lines 526-537).

Comment 14: Section 3.6 is irrelevant to the study. Presenting more information does not necessarily improve manuscript quality.

Response: Thank you very much for pointing out this. Though comparable performance was achieved on tomato plants, it would be great if the proposed pipeline can be implemented on other plants. Thus, the generalization ability of 3DPhenoMVS was testied on other plants, and the potential has been demonstrated in this study.

Comment 15: Line 439, how do you know laser point cloud accuracy is the highest?

Response: Thank you very much for pointing out these. Lidar and structured light scanner are two common active 3D scanning sensors. Their precision is relatively high compared to other methods. This conclusion can be proved by the following two points:

  1. From the principle of the two methods, structured light is a system that calculates the position, depth and other information to restore the entire object. Lidar is a system that emits a laser beam to detect the position of the target point to restore the object. Their principles are similar, and the information of the obtained point cloud comes directly from the sensor, which is not easily affected by the external environment and human factors.
  2. From the experimental results, Wang et al. compared the point clouds obtained by 3D scanning and three representative 3D reconstruction methods [9] (Reference 9 in the literature). The experimental results show that 3D scanning can obtain high precision and reliable points cloud, which proves that the Lidar and structured light are widely used 3D scanning methods with relatively high accuracy.

However, the existing low-cost 3D scanning sensors are only suitable for high-scale and low-detail application scenarios such as unmanned driving. They are not suitable for accurate phenotypic trait estimation of individual plant at organ levels. Therefore, to demonstrate the performance of the proposed pipeline, point clouds generated by LiDAR and structured light sensor were chosen as the reference data.

Based on the discussion above, we revised the description as follows, and also add the reference literature. Thanks again for the comments.

“As shown in Figure 4a-4h, the accuracy of point clouds generated by LiDAR or structured light sensor is the highest [9]”

Again, thanks the editor and reviewer for their constructive comments and suggestions. We really appreciate that.

Round 2

Reviewer 3 Report

I am unsatisfied with some of the author's response. See below for more details. I need to see significant changes in the next revised manuscript.  
  • "Among all the 19 traits estimated, true values for plant volume would be acquired by water displacement method, and accurate leaf area would be measured by leaf area meter. Both the methods were destructive and would cause destruction to tomato plants, which is not consistent with our research object, since we are focusing on the whole plant growth stages of tomato plants."
This is not a valid reason for not evaluating plant volume. You clearly claimed in the abstract: "we proposed a 3D phenotyping pipeline, 3DPhenoMVS, to calculate and analyze 19 phenotypic traits of tomato plants covering the whole life cycle." I understand that your pipeline has the potential to estimate plant volume, however, it is irresponsible and unprofessional to make such claims without conducting actual evaluation. I suggest simply reducing your 19 traits to 18 if you would not like to do additional experiments. Also I would like to see a summary of your response to my comment 1 so that readers can understand why not all 19/18 traits have the scatter plots presented.
  • Being a lidar researcher for years, I pointed out in comment 3 that low-cost lidars are currently available, from $100 2D lidars to $600 3D lidars. You gave an unnecessarily lengthy response full of flaws. First of all, I am unsure where you learned the term "dynamic lidar", because this is simply not such a concept. Velarray M1600 is not even available on the market and I'm not sure how you learned the price. You said these lidars "are mainly used for large/medium scale scenarios". Why can't they be used for scanning small objects? You said these lidars' "accuracy is not high". By what standard? +/-2 cm is high? Also what is stopping you from collecting more lidar frames to generating dense point clouds using such lidars?
  • Update your response to comment 5 in the manuscript.
  • You entirely avoided addressing my comment 8 so I'll repeat myself one more time below. Before that I would like to make it clear that I do not appreciate you kept using your focus is on photogrammetry as an excuse to not provide important methodological details. 
Section 3.2 messy structure. Most of the content should be in the method and material section. Lidar and structured light scanner data collection methods are not described in detail at all. What are the models of the sensors? How did you register your lidar frames (I need to see your registration algorithm or your GPS/IMU model and georeferencing equations)? Additionally, when you use lidar and structured light scanner point clouds as ground truth, how do you know these point clouds themselves are accurate?
  • Update your response to comment 13 in the manuscript. Also you did not address one of my questions: what about the rest traits regarding light intensity, temperature and humidity?
  • In your response to comment 15, you said lidars "are not suitable for accurate phenotypic trait estimation of individual plant at organ levels". This is a false claim since numerous studies involving lidars and plant organs exist. Also, your response did not address my comment 15 at all, so I will ask one more time: how do you know laser point cloud accuracy is the highest without using quantitative metrics? Point cloud accuracy largely depends on registration accuracy, not sensor measurement accuracy.

Author Response

Response to reviewer#3

We really appreciate for your constructive comments and suggestions of this manuscript. We addressed each comment in turn below. The answers and revisions to each comment have been highlighted using blue and red fonts to make it easily visible. Should there been any other suggestions and comments, please feel free to contact us.

Again, please accept our endless gratitude for improving our manuscript.

Comment 1: "Among all the 19 traits estimated, true values for plant volume would be acquired by water displacement method, and accurate leaf area would be measured by leaf area meter. Both the methods were destructive and would cause destruction to tomato plants, which is not consistent with our research object, since we are focusing on the whole plant growth stages of tomato plants."

This is not a valid reason for not evaluating plant volume. You clearly claimed in the abstract: "we proposed a 3D phenotyping pipeline, 3DPhenoMVS, to calculate and analyze 19 phenotypic traits of tomato plants covering the whole life cycle." I understand that your pipeline has the potential to estimate plant volume, however, it is irresponsible and unprofessional to make such claims without conducting actual evaluation. I suggest simply reducing your 19 traits to 18 if you would not like to do additional experiments. Also I would like to see a summary of your response to my comment 1 so that readers can understand why not all 19/18 traits have the scatter plots presented.

Response: Thank you very much for pointing out this. We totally agree with you and have adopted your suggestion to reduce the number of traits from 19 to 17 by removing plant volume and leaf area. Among all the phenotypic traits, 6 of them were used for accuracy evaluation. We have made corrections in the abstract subsection as follows.

Line 13-16. “…, we proposed a 3D phenotyping pipeline, 3DPhenoMVS, to calculate 17 phenotypic traits of tomato plants covering the whole life cycle. Among all the phenotypic traits, six of them were used for accuracy evaluation because the true values can be generated by manual measurements, and the results showed that the R2 values between the phenotypic traits and the manual ones ranged from 0.72 to 0.97.”

At the same time, we also include the description answering your comment 1 in the revised manuscript in subsection 3.3 as shown by line 416-419.

Line 416-419. “Since true values of 6 phenotypic traits including plant height, stem length, internode length, transverse diameter, vertical diameter, and node number can be measured manually and counted, the accuracy estimation was conducted on them, and the comparative results are shown in Figure 5a-5f.”

Comment 2: Being a lidar researcher for years, I pointed out in comment 3 that low-cost lidars are currently available, from $100 2D lidars to $600 3D lidars. You gave an unnecessarily lengthy response full of flaws. First of all, I am unsure where you learned the term "dynamic lidar", because this is simply not such a concept. Velarray M1600 is not even available on the market and I'm not sure how you learned the price. You said these lidars "are mainly used for large/medium scale scenarios". Why can't they be used for scanning small objects? You said these lidars' "accuracy is not high". By what standard? +/-2 cm is high? Also what is stopping you from collecting more lidar frames to generating dense point clouds using such lidars?

Response: First of all, we are really sorry for the wrong claims and unprofessional words in the manuscript.

“First of all, I am unsure where you learned the term "dynamic lidar", because this is simply not such a concept. “

Answer: Thank you very much for correcting us. We have investigated the principles of LiDAR systems, and will not use the term “dynamic LiDAR” in the future.

“Velarray M1600 is not even available on the market and I'm not sure how you learned the price.Velarray M1600 is not even available on the market and I'm not sure how you learned the price. You said these lidars "are mainly used for large/medium scale scenarios. Why can't they be used for scanning small objects?".

Answer: According to the information released by Beidouxingtong, the Chinese agent of Velarray, the M1600 is suitable for low-speed unmanned logistics vehicles and automatic driving scenarios. Based on that, we concluded that they cannot be sued for small object scanning. We totally agree with you that the LiDAR accuracy was estimated by the registration accuracy but not the sensor measurement accuracy. If dense point clouds were wanted regarding small sized objects, enough scans and advanced registration algorithms would help. We have corrected the description in the main text.

“You said these lidars' "accuracy is not high". By what standard? +/-2 cm is high?

Answer: We realized our mistakes that we have made on the discussion of LiDAR systems. We should not give a simple conclusion that “accuracy is not high.”, which is unfounded. We have corrected the description in the main text.

Also what is stopping you from collecting more lidar frames to generating dense point clouds using such lidars?”

Answer: Since we were trying to investigate the 3D phenotyping technologies by using image based data. Thus, we use the point clouds collected other two active sensors as the reference data. In addition, we supplemented the necessities of investigation and deployment of low-cost LiDAR systems for plant phenotyping in the outlook and perspective.

Line 608-610. “Except the image based low-cost phenotyping pipeline, the applications of new emerged low cost LiDAR systems in phenotyping needs to be further explored.”

Comment 3: Update your response to comment 5 in the manuscript.

Response: Thank you very much for pointing out this. We have revised the description in Subsection 2.3.

Line 221-225. “Theoretically, all the images including the upper and lower parts of tomato plants at the later growth stages can be used for 3D point cloud generation simultaneously by inputting all of them. In practice, due to the limitation of our computing power, if the number of images was too large, the computer would crash. Thus, the upper part and lower part were reconstructed separately.”   

Comment 4: You entirely avoided addressing my comment 8 so I'll repeat myself one more time below. Before that I would like to make it clear that I do not appreciate you kept using your focus is on photogrammetry as an excuse to not provide important methodological details.

Section 3.2 messy structure. Most of the content should be in the method and material section. Lidar and structured light scanner data collection methods are not described in detail at all. What are the models of the sensors? How did you register your lidar frames (I need to see your registration algorithm or your GPS/IMU model and georeferencing equations)? Additionally, when you use lidar and structured light scanner point clouds as ground truth, how do you know these point clouds themselves are accurate?

Response: We sincerely apologize for having overlooked these issues. We have put too much emphasis on the agricultural application without considering the registration methods themselves.

“What are the models of the sensors?”

Answer: Thank you again for reminding us. In this study, we used a LiDAR sensor and a structured light scanner to collect point clouds as reference data. With respect to the LiDAR system, a FARO Laser Scanner (FARO Focus S70, FARO Corporation, America), following the indirect time of flight principle, Regarding the structured light scanner, 3D structured light scanner (Reeyee Pro, Wiiboox Corporation, China) was adopted for data acquisition of one plant in the indoor scenario. We have added the description in the main text.

“How did you register your lidar frames (I need to see your registration algorithm or your GPS/IMU model and georeferencing equations)?”

Answer: Theoretically, multiple LiDAR frames need to be registered and aligned together to produce a complete point cloud by using iterative closest point (ICP) algorithms or the improved ones. In our study, the registration and alignment procedures were completed by commercial software directly. We think that ICP algorithms might be adopted by the companies, however, since the software are not open source, that’s why we did not provide more details for the registration algorithms.

Additionally, when you use lidar and structured light scanner point clouds as ground truth, how do you know these point clouds themselves are accurate?

Answer: Thank you very much for the remind. We supplemented the description of the accuracy of point clouds generated from the LiDAR and structured light sensor, and they are around 3.40mm and 0.15mm respectively. Based on the accuracy of the results, we concluded that the point clouds generated by the two sensors could be used as ground truth for reference.

According to your suggestion, the data collection of 3D point cloud from active sensors were complemented in subsection 2.8.

Line 318-341. “To evaluate the performance of the reconstructed point clouds from images, LiDAR and structured light scanner were used to collect point clouds as reference data for comparison. As to the LiDAR system, a FARO Laser Scanner (FARO Focus S70, FARO Corporation, America), following the indirect time of flight principle, was used for data collection in the greenhouse. A group of 3 target balls with a diameter of 0.15m were placed in the scenario, and the sensor was mounted on a tripod at a consistent height. A total of 6 LiDAR frames covering the area of 10m x 1.5m were collected. Two LiDAR frames were registered and aligned together first with the aid of the target balls to produce one set of point cloud consisting of the first two LiDAR frames. Later, the other LiDAR frames were registered and aligned with the produced point clouds following the same procedures one by one. All the procedures were implemented by using the commercial software packages SCENE provided by FARO. A complete point cloud was generated, with an average error of 3.4mm, which depicted the distance between the points from two LiDAR scans.

In regard to the structured light sensor, a 3D structured light scanner (Reeyee Pro, Wiiboox Corporation, China), leveraging white light into accurate measurements, was adopted for data acquisition of individual plant or tomato fruit in the indoor scenario. One tomato plant or fruit was placed on a rotary table with fluorescent targets controlled by a step motor. The rotation angle was set at 20°, and the commercial software Reeyee-Pro-v2 managed the data collection and point cloud generation in an automatic fashion. A complete point cloud of individual plant was created with error around 0.15 mm, and tomato fruit with error around 0.12mm.

Based on the discussion above, both the accuracy of the point clouds generated by a LiDAR sensor and a structured light sensor were quite high, and thus, they were used as reference data for later accuracy evaluation.”

Comment 5: Update your response to comment 13 in the manuscript. Also you did not address one of my questions: what about the rest traits regarding light intensity, temperature and humidity?

Response: Thank you very much for pointing out this. We have revised the description in subsection 3.4.

What about the rest traits regarding light intensity, temperature and humidity?”

Answer: All the phenotypic trait parameters were analyzed in combination with the environmental parameters. Among them, stem length, plant height, and plant volume show great difference as the environmental parameters varied, and the other parameters did not show such trends. Thus, we used three of them to analyze the influence of the environmental differences to the growth of tomato plants.

The revised version is as follows.

Lines 438-447: “We divided the locations of each environmental parameter into two categories according to the environmental difference distribution map (Figure 2b), such as locations with high intensity and low intensity, locations with high humidity and low humidity, and locations with high temperature and low temperature. The tomato samples were classified into two categories by their locations. All the phenotypic traits were analyzed in combination with the environmental parameters. Among them, stem length, plant height, and plant volume show great difference as the environmental parameters varied, and the other parameters did not present such trends. Thus, we used three of them to analyze the influence of the environmental differences to the growth of tomato plants.”

Comment 6: In your response to comment 15, you said lidars "are not suitable for accurate phenotypic trait estimation of individual plant at organ levels". This is a false claim since numerous studies involving lidars and plant organs exist. Also, your response did not address my comment 15 at all, so I will ask one more time: how do you know laser point cloud accuracy is the highest without using quantitative metrics? Point cloud accuracy largely depends on registration accuracy, not sensor measurement accuracy.

Response: Thank you very much for pointing out this. We are really sorry for the reckless remarks. In the field of phenotyping, there are numerous studies using LiDAR for individual plant or plant organ segmentation, such as maize, cotton and other plants. We apologize again for the mistakes.

Wang, Y.J.; Wen, W.L.; Wu, S.; Wang, C.Y.; Yu, Z.T.; Guo, X.Y.; Zhao, C.J. Maize Plant Phenotyping: Comparing 3D Laser Scanning, Multi-View Stereo Reconstruction, and 3D Digitizing Estimates. Remote Sensing 2018, 11, 63.

Jin, S.C.; Su, Y.J.; Wu, F.F.; Pang, S.X.; Gao, S.; Hu, T.Y.; Liu, J.; Guo, Q.H. Stem-Leaf Segmentation and Phenotypic Trait Extraction of Individual Maize Using Terrestrial LiDAR Data. IEEE Transactions on Geoscience and Remote Sensing 2019, 57, 1336-1346.

Sun S.P.; Li C.Y.; Chee W.P.; Paterson A.H.; Meng C.; Zhang J.Y.; Ma P.; Robertson J.S. High Resolution 3D terrestrial LiDAR for Cotton Plant Main Stalk and Node Detection. Computers and Electronics in Agriculture 2021, 187, 106276.

How do you know laser point cloud accuracy is the highest without using quantitative metrics? Point cloud accuracy largely depends on registration accuracy, not sensor measurement accuracy.

Answer: We are really sorry again for not addressing this problem. We supplemented the description of the accuracy of point clouds generated from the LiDAR and structured light sensor, and they are around 3.40mm and 0.15mm respectively. Based on the accuracy of the results, we concluded that the point clouds generated by the two sensors could be used as ground truth for reference.

The detailed information was shown in subsection 2.8.

Line 318-341. “To evaluate the performance of the reconstructed point clouds from images, LiDAR and structured light scanner were used to collect point clouds as reference data for comparison. As to the LiDAR system, a FARO Laser Scanner (FARO Focus S70, FARO Corporation, America), following the indirect time of flight principle, was used for data collection in the greenhouse. A group of 3 target balls with a diameter of 0.15m were placed in the scenario, and the sensor was mounted on a tripod at a consistent height. A total of 6 LiDAR frames covering the area of 10m x 1.5m were collected. Two LiDAR frames were registered and aligned together first with the aid of the target balls to produce one set of point cloud consisting of the first two LiDAR frames. Later, the other LiDAR frames were registered and aligned with the produced point clouds following the same procedures one by one. All the procedures were implemented by using the commercial software packages SCENE provided by FARO. A complete point cloud was generated, with an average error of 3.4mm, which depicted the distance between the points from two LiDAR scans.

In regard to the structured light sensor, a 3D structured light scanner (Reeyee Pro, Wiiboox Corporation, China), leveraging white light into accurate measurements, was adopted for data acquisition of individual plant or tomato fruit in the indoor scenario. One tomato plant or fruit was placed on a rotary table with fluorescent targets controlled by a step motor. The rotation angle was set at 20°, and the commercial software Reeyee-Pro-v2 managed the data collection and point cloud generation in an automatic fashion. A complete point cloud of individual plant was created with error around 0.15 mm, and tomato fruit with error around 0.12mm.

Based on the discussion above, both the accuracy of the point clouds generated by a LiDAR sensor and a structured light sensor were quite high, and thus, they were used as reference data for later accuracy evaluation.”
